# Elevated Likelihood of Infectious Complications Related to Oral Mucositis After Hematopoietic Stem Cell Transplantation: A Systematic Review and Meta-Analysis of Outcomes and Risk Factors

**DOI:** 10.3390/cancers17162657

**Published:** 2025-08-14

**Authors:** Susan Eichhorn, Lauryn Rudin, Chidambaram Ramasamy, Ridham Varsani, Parikshit Padhi, Nour Nassour, Kapil Meleveedu, Joel B. Epstein, Benjamin Semegran, Roberto Pili, Poolakkad S. Satheeshkumar

**Affiliations:** 1Boston Medical Center, Boston, MA 02118, USA; susan.eichhorn@bmc.org; 2Jacobs School of Medicine and Biomedical Sciences, University at Buffalo, Buffalo, NY 14203, USA; laurynru@buffalo.edu (L.R.); nournass@buffalo.edu (N.N.); bssemegr@buffalo.edu (B.S.); 3Carol and Ray Neag Comprehensive Cancer Center, University of Connecticut Health Center, Farmington, CT 06030, USA; cramasamy@uchc.edu (C.R.); kmeleveedu@uchc.edu (K.M.); 4School of Dental Medicine, University at Buffalo, Buffalo, NY 14215, USA; ridhamva@buffalo.edu; 5Kaleida Health Infusion Center, Department of Medicine, Division of Hematology and Oncology, University at Buffalo, Buffalo, NY 14221, USA; ppadhi@buffalo.edu; 6City of Hope National Cancer Center, Duarte, CA 91010, USA; jepstein@coh.org; 7Department of Medicine, Division of Hematology and Oncology, University at Buffalo, Buffalo, NY 14203, USA; rpili@buffalo.edu

**Keywords:** oral mucositis, hemopoietic stem cell transplants, risk factors, outcomes, systematic reviews, meta-analysis

## Abstract

Oral mucositis (OM) is a frequent, debilitating side effect of hematopoietic stem cell transplantation (HSCT), significantly impacting patient outcomes and quality of life. This systematic review and meta-analysis attempted to establish risk factors for OM and its association with infectious complications in recipients of HSCT. Thirty-four studies were conducted, and high-intensity conditioning, administration of methotrexate, female sex, longer neutropenia/neutrophil engraftment, reactivation of HSV-1 infection, and renal impairment appeared as significant risk factors for OM. This meta-analysis demonstrated that patients suffering from OM had nearly four times the risk of developing infections compared with non-OM patients. These findings indicate the importance of early OM diagnosis and OM-prevention strategies in restricting severe complications in immunocompromised patients receiving HSCT.

## 1. Introduction

Oral and oropharyngeal mucositis (OM) is a well-known consequence of cancer therapy, particularly hematopoietic stem cell transplantation (HSCT), and is associated with pain, nutritional deficiencies, dehydration, respiratory distress, treatment delays, and an increased risk of febrile neutropenia and systemic infections [1,2,3]. The preparatory or conditioning regimen for HSCT, comprising high-dose chemotherapy with or without total body irradiation (TBI), is designed to eradicate residual malignant cells and immunosuppress the host to allow engraftment. It can also harm the mucosal layer and other highly replicative cells in the body [4]. Following the infusion of stem cells, patients enter a prolonged aplastic period where granulocyte recovery is delayed, leaving them susceptible to infection and mucosal injury. Engraftment is marked by the return of neutrophil counts, typically around day +14 to +21, though this timing can vary depending on the graft source and patient-specific factors [4]. Hospitalization during this phase is often prolonged, particularly in patients with delayed engraftment, graft-versus-host disease (GVHD), or infectious complications [4]. Notably, both pediatric and adult populations undergo HSCT, but elderly recipients, whose numbers have risen significantly in recent years alongside rising rates of HSCT treatments, are often at increased risk due to comorbidities and reduced mucosal regenerative capacity [5,6].

According to recent studies, 80% of patients undergoing HSCT are expected to develop OM, 35–75% of those with autologous transplants, and 75–100% of those with allogeneic transplants [2,7,8]. These rates are significantly higher than in patients receiving chemotherapy alone [2]. Not only is OM prevalent, but it tends to occur in advanced forms, with 42–60% of HSCT patients developing severe OM [2,9]. This may lead to chemotherapy dose reduction or discontinuation, with negative consequences for patient outcomes [2,9]. The oral cavity, particularly when compromised by mucositis, serves as a reservoir for pathogenic organisms capable of translocating into systemic circulation, especially in immunocompromised populations such as hematologic malignancy patients [10,11]. Studies have shown that bloodstream infections are estimated to affect about 5–10% of autologous and 20–30% of allogeneic HSCT recipients, though this figure can change on a center-by-center and patient-by-patient basis [12]. For example, some recent studies have even shown rates as high as 77% for autologous transplant and 48% for allogeneic transplant [13,14]. Given that OM is a predictable, dose-limiting toxicity in patients receiving high-dose chemotherapy or HSCT [1,2], identifying its role in systemic infection is critical for developing preventative strategies.

Unfortunately, methods to prevent OM remain limited, and most commonly involve oral cryotherapy, growth factors, and benzydamine. Once an individual develops OM, there is no cure, and treatment is focused on managing pain with analgesics and low-level laser therapy, if available [1]. Poor oral health [15,16], smoking [15,17], genetic polymorphisms [18], and high-intensity conditioning regimens [2,15,17,19] are considered risk factors for OM in HSCT recipients. However, the majority of information is collected from randomized controlled trials (RCTs), in which females and minorities are underrepresented [20,21]; hence, compiling results from observational studies to obtain real-world evidence is needed at any cost, and further examining oral-systemic relationship is crucial to preventing poor outcomes in this cohort of patients [22]. A 2015 study by Chaudhry et al. analyzed the risk of OM in allogeneic HSCT patients in terms of regimen intensity and GHVD prophylaxis with or without methotrexate [19]. A subsequent study by Wardill et al. in 2020 looked more broadly at OM risk factors in patients receiving cancer therapy, not only those undergoing HSCT [17]. It assessed not only treatment modalities but patient characteristics, including genetic polymorphisms and demographic/lifestyle elements. We look to expand these areas of knowledge by further examining patient and clinical risk factors associated with OM in HSCT patients, as well as infectious outcomes. In doing so, we hope to identify higher-risk individuals and ultimately reduce incidence through earlier detection and employment of prophylactic strategies. Our objectives are thus twofold:
(1)Identify risk factors associated with OM across HSCT modalities from multivariate analyses.(2)Quantify the risk of infectious complications associated with OM.

## 2. Methods

### 2.1. Search and Screening Protocol

A systematic literature review was conducted following the PRISMA statement (Figure 1 and Figure 2) [23]. The search was performed in PubMed and EBSCO and included publications from inception until 31 March 2024. Search terms were selected from the Medical Subject Headings (MeSH) database to maximize the retrieval of relevant publications (Appendix A). Title and abstract were independently screened by three investigators (SE, LR, and PSS) and further confirmed by others (LR, RV, and RC). Disagreement regarding eligibility was discussed and resolved.

Studies identified in the literature search were included or excluded from the analysis based on the following criteria.

Inclusion and exclusion criteria for risk factor analysis.


**Inclusion Criteria for Risk Analysis**

**Exclusion Criteria for Risk Analysis**

The study population consisted of adult patients (≥18 years old) undergoing HSCT.The study reported results from a multivariate analysis that adjusted for identified confounders.Reported quantitative data about infectious complications following OM in patients undergoing HSCT treatment.Quantitative data about risk factors associated with OM was reported as an outcome in patients undergoing HSCT treatment.

The following study or article designs include case reports, case series, systematic reviews, meta-analyses, qualitative studies of knowledge, attitude, and perspective (KAP) nature, protocol designs, validation studies, animal studies, and guidelines.The study population consisted of patients undergoing HSCT and associated conditioning regimen as therapy for non-cancer conditions, including autoimmune diseases.The study was not published in English.The study lacked a clearly defined intervention or outcome.The study did not include data from a multivariate analysis.The study reported gastrointestinal mucositis as the only risk factor of interest.


Inclusion and exclusion criteria for analysis of infectious complications.


**Inclusion Criteria for Infection Outcomes**

**Exclusion Criteria for Infection Outcomes**

The study population consisted of adult patients (≥18 years old) undergoing HSCT.The study reported results from observational and randomized control studies.Reported quantitative data about infectious complications following OM in patients undergoing HSCT treatment.Infectious complications included systemic viral, bacterial, or fungal infections and other sepsis-related outcomes.

The following study or article designs: case reports, case series, systematic reviews, meta-analyses, qualitative studies of knowledge, attitude, and perspective (KAP) nature, protocol designs, validation studies, animal studies, and guidelines.The study population consisted of patients undergoing HSCT and associated conditioning regimen as therapy for non-cancer conditions, including autoimmune diseases.The study was not published in English.The study lacked a clearly defined intervention and/or outcome.The study did not include data from a multivariate analysis.The study reported on gastrointestinal mucositis as the only outcome of interest.


### 2.2. Data Extraction and Assessment of Studies

We focused on the two aforementioned objectives, utilizing separate population, intervention, comparison, and outcome (PICO) questionnaires to modulate our inclusion and exclusion criteria for studies included in the review and meta-analysis. For the risk factor analysis, the population was HSCT patients receiving high-dose chemotherapy, comparing mucositis vs. non-mucositis cases and their associated risk factors. Looking specifically at infectious disease complications, we again evaluated HSCT patients and compared mucositis vs. non-mucositis outcomes, looking for any viral, fungal, and/or bacterial infections between the two populations.

Utilizing the generated inclusion vs. exclusion criteria, SE examined the body of studies yielded by our search terms (Appendix A), which consisted of MeSH terms and Boolean frameworks for searching PubMed and EBSCO. Further data extraction was performed by NN, RV, LR, and CR, followed by validation by RP and JE. The studies, findings, and biases were validated by PSS and JE utilizing different scales; for non-randomized, observational studies, we used the Newcastle-Ottawa scale, and for randomized trials, we used the Cochrane Collaboration’s risk of bias tool, with further validation performed by LR, CR, and RV. The Newcastle-Ottawa Scale evaluated potential bias in the following three domains: selection of participants, comparability, and exposure assessment [24]. To be included in the review, a study had to achieve a minimum score of 6 out of a possible 9 points. Cochrane’s risk of bias tool evaluated selection of participants, performance, detection, attrition, reporting, and any other identified bias [25]. Included studies had green ratings for at least three bias criteria and no more than two red ratings for bias criteria. We examined the multivariate analysis of the observational studies to derive our risk factors, and those examined with a multivariate analysis in the adjustment analysis served to adjust for the confounding factors. It was during this phase of full text screening, after studies were screened by title/abstract and ability to be retrieved, that study identification for risk factor analysis and infectious complication analysis diverged.

### 2.3. Statistical Analysis

A meta-analysis was performed comparing the incidence of infectious complications between patients with OM and patients without. A DerSimonian and Laird random effects model was utilized for analysis. Tests of heterogeneity were conducted with an I2 score greater than 50% or a *p*-value less than 0.05 considered evidence of heterogeneity. Statistical analysis was performed using R Studio, Version 4.3, Vienna, Austria.

## 3. Results

### 3.1. Search Results

Our initial search identified 1677 articles, of which 34 were included in our study. Of those 34 studies, 4 were included in the meta-analysis assessing the relationship between OM and infectious complications (Table 1 and Table 2) [26,27,28,29,30,31,32,33,34,35,36,37,38,39,40,41,42,43,44,45,46,47,48,49,50,51,52,53,54,55,56,57,58,59,60]. A 2022 retrospective cohort study assessed for inclusion was ultimately not included in the final meta-analysis due to the low frequency of events in the control group, which can significantly impact a study’s validity and ability to draw reliable conclusions [27]. Thirty studies were included in the qualitative assessment of risk factors in the development of OM (Table 2) [18,31,32,33,34,35,36,37,38,39,40,41,42,43,44,45,46,47,48,49,50,51,52,53,54,55,56,57,58,59,60] Two different study designs were included in the meta-analysis: two case-control studies and two cohort studies. Two different study designs were included in the risk factor review: twenty-seven cohort studies and three randomized controlled trials. All studies were found to meet minimum quality standards for inclusion in the study (Appendix A). Both case-control studies in the meta-analysis scored an 8/9 with the Newcastle-Ottawa Scale. The two cohort studies included in the meta-analysis scored 7/9 and 9/9 with the Newcastle-Ottawa Scale. Regarding the 27 cohort studies included in the risk factor analysis, 4 studies scored 6/9, 13 studies scored 7/9, and 10 studies scored 8/9.

### 3.2. Meta-Analysis of Effect of Oral Mucositis on Developing Infectious Complications

Our meta-analysis included four studies with a total of 611 patients in our analysis [26,28,29,30]. Data from multivariate analyses was available in Anaissie et al., 2004 and Levallee et al., 2016 [26,28]. Mikulska et al., 2010 only provided data from a univariate analysis, but within their study, they determined a statistically significant relationship between OM and enterococcal bacteremia in multivariate analysis (OR 9.04 [1.97–41.52], *p* = 0.018) [29]. We determined this to be sufficient for inclusion in our study. Santos et al., 2012 only provided data from a univariate analysis as well, and in a multivariate analysis between all grades of OM and infection, they did not find a statistically significant relationship (OR 2.21 [0.98–4.94], *p* = 0.054) [30]. The authors noted that severe OM was associated with more frequent infections [30]. We concluded that if severe OM was used in multivariate analysis, there likely would have been a statistically significant relationship, and for that reason, grades 0-I OM were included in the control group and grades II-IV OM as exposure of interest. The findings demonstrated that patients with OM have an increased risk of developing infectious complications compared to those without OM. The analysis revealed a pooled odds ratio of 3.84 (95% CI: 2.51–5.86, I^2^ = 0%, tau^2^ = 0, *p* value_het_ = 0.96) (Figure 3). Infectious complications identified in included studies were serious systemic complications of respiratory syncytial virus (RSV), assays positive for *Clostridium difficile* toxin and antigen, and blood cultures positive for *Enterococcus* and other bacterial and fungal infections [26,28,29,30]. RSV infection-related complications included upper respiratory infections, lower respiratory infections, pneumonia, renal failure, and subsequent infections [26].

### 3.3. Oral Mucositis Risk Factors Across HSCT Recipients

Our study identified several risk factors, which were organized into one of the following four categories: baseline patient characteristics, laboratory results, cancer treatment and conditioning regimens, and use of OM prophylaxis (Table 3). Fourteen studies identified baseline patient characteristics as risk factors for the development of OM. Potential risk factors include age < 40 years, female sex, presence of HSV-1, reduced renal function, malnutrition, poor functional status, poor immune status, and several genetic factors [18,33,36,38,39,40,43,46,47,49,50,53,54,55]. Laboratory results, including higher ferritin level, longer duration of neutropenia, an increase in inflammatory cytokines such as plasma and salivary interleukin-6, and presence of specific bacterial and fungal species in oral microbiota, demonstrated increased risk of OM in six studies [31,36,40,42,52,57]. Sixteen studies reported features of cancer treatment and HSCT conditioning regimens that were associated with increased risk of OM. Use of myeloablative conditioning, use of methotrexate, higher doses of melphalan, higher doses of carmustine, conditioning with the BEAM regimen or busulphan, use of bendamustine, use of TBI, and use of multiple treatment lines were associated with increased risk of OM [32,33,34,35,37,38,41,45,46,47,48,51,57,59,60]. Regimens used for non-Hodgkin’s lymphoma (NHL) also demonstrated a higher risk of OM compared to Hodgkin lymphoma (HL) treatment protocols [50].

In three studies, protective factors in treatment, conditioning regimens, and laboratory results were identified—autologous HSCT compared to allogeneic HSCT, use of reduced intensity conditioning, and hypomagnesemia [35,46,56]. Five studies also evaluated the risk of OM with different prophylactic strategies. Folinic acid and cryotherapy as prophylaxis were found to be protective against OM, and use of any prophylaxis was associated with decreased risk of OM [32,37,50,52,53].

## 4. Discussion

OM is an established complication of cancer therapy, including HSCT with chemotherapeutic and radiation conditioning regimens [1,2]. Unfortunately, a gap in knowledge about factors that may increase a patient’s risk of developing OM remains [23]. This analysis identified several potential risk factors for OM in HSCT recipients with cancer from 30 multivariate analyses across various categories, including baseline patient characteristics, laboratory results, and cancer treatment and conditioning regimens. Additionally, our understanding of the effect of OM on patient outcomes remains limited. Findings here demonstrated a significantly increased risk of infectious complications in HSCT recipients with cancer who develop OM.

Previous research has focused on chemotherapy and radiation-induced OM [1]. Chaudhry et al. (2016) only identified risk factors from allogeneic HSCT recipients [19]. Wardill et al. (2020) looked at risk factors across both HSCT modalities, but the review included data from univariate analyses [17]. Our risk factor review included factors from multivariate analyses across all stem cell transplantation modalities. Our search did not identify any independent review of risk factors and incidence of OM among autologous HSCT recipients. The association between the incidence of OM and subsequent infectious complications in cancer patients who have received HSCT has significant clinical implications for treating this vulnerable population. One possible mechanism for increased infection risk in patients with OM is that the oral ulcers provide opportunities for local infection with microbes, such as HSV-1, *P. gingivalis*, and *Candida*, which ultimately leads to systemic sepsis [1]. This is consistent with studies included in this paper, which found that higher viral loads of HSV-1 and the presence of non-albicans *Candida* species and bacteria, such as *P. gingivalis*, in the oral mucosa were associated with increased incidence of ulcerative OM [39,42,44].

This result may be even more pronounced in patients with ulcerative OM, and the severity of OM may be directly correlated with the likelihood of subsequent infection. Santos et al. (2012) did not find a statistically significant relationship between OM and bacterial and *Candida* infections in autologous HSCT recipients; however, they concluded that patients with severe OM experienced higher rates of infection [30]. For this reason, in our analysis, we only included data from patients with grades II-IV OM [30]. Similar results were demonstrated in the relationship between grade III/IV OM and enterococcal bacteremia in allogeneic HSCT recipients [29]. This consideration further demonstrates that identifying patients at higher risk of developing OM may allow for earlier intervention that may reduce not only the severity of OM but also the subsequent risk of infection and other serious complications.

Infections following OM present many challenges and potentially life-threatening consequences for patients undergoing HSCT [1,2]. Deveci et al. (2022) showed that the presence of OM in bone marrow recipients was an independent risk factor for typhlitis, a common and life-threatening infectious complication in immunocompromised patients [27]. One study analyzing patients who developed mucosal barrier injury–laboratory confirmed bloodstream infections (MBI-LCBI) following allogeneic HSCT, determined that patients with infection had significantly higher one-year mortality rates than those without infection (HR 1.81 [99% CI: 1.56–2.12]) [61]. The authors also found use of myeloablative conditioning (MAC) to be a significant risk factor for the development of MBI-LCBI, which is consistent with this study’s conclusion that MAC is an independent risk factor for OM [61]. Another study found that infections resulting from ulcerative OM were associated with a greater burden of illness, as defined as longer length of hospital stay, greater cost of treatment, and discharge to non-home facilities, compared to cancer patients who did not have infections with OM [62]. Mucositis-related infections negatively impact not only the health and safety of the patients but also the utilization of hospital resources. If OM is one of the links between HSCT therapy and infection, it should be a priority to investigate risk factors and identify high-risk patients to minimize the incidence of burdensome infectious complications. Our study identified multiple risk factors associated with the development and severity of OM in patients undergoing HSCT. Unlike Wardill et al. (2020) [17], our analysis includes only studies that reported multivariate analyses, strengthening the association between identified risk factors and OM by adjusting for potential confounders.

Treatment-related factors, such as high-intensity chemotherapy used in MAC and radiation conditioning regimens, are risk factors for OM. This is consistent with previous studies [2,15,17,19]. Chemotherapy agents involved in high-intensity, myeloablative regimens include cytarabine, high-dose 5-fluorouracil (5-FU), alkylating agents, and platinum-based compounds, all of which have been associated with a high incidence of OM [32,33,34,35,37,38,41,45,47,57,59,60]. Duration of these treatment courses and subsequent neutropenia are also correlated with increased OM severity [2,36,40,52]. Prolonged neutropenia delays mucosal healing and increases the risk of secondary infections, which can further exacerbate tissue breakdown [11]. Additionally, extended exposure to cytotoxic agents during conditioning regimens intensifies cumulative epithelial injury, compounding the severity and duration of mucositis [11]. Certain genetic polymorphisms involved in drug metabolism are associated with increased risk of OM development [17]. Physicians may consider using prophylactic agents in patients who will require higher intensity or myeloablative regimens.

Prevention of OM is limited as few agents have been shown to positively impact the level of risk; however, physicians may still consider providing these treatments to patients at greater risk of OM with the hope of reducing even the severity of OM. Although oral cryotherapy was not found to effectively reduce the risk of OM in patients undergoing allogeneic HSCT, it was associated with positive outcomes in patients undergoing autologous HSCT [32,37,63]. Palifermin is another agent that may have some meaningful benefit as a prophylactic therapy for reducing the severity of OM [48]. Additionally, if a patient has multiple risk factors for developing OM, physicians may consider using reduced intensity conditioning regimens, which have been associated with a lower risk of OM [46,51]. With these findings, providers may possess sufficient knowledge to accurately balance the risk of incomplete remission in cancers requiring HSCT with the risk of potentially life-threatening infections that may occur due to the development of OM.

Methotrexate is an antimetabolite agent widely used to prevent GVHD, a life-threatening consequence of allogeneic HSCT in which donor cells initiate an immune response against host tissues [58,64]. While powerful against GVHD, our study suggests that its use in stem cell transplant recipients may increase the risk of OM. Several studies have shown that within cohorts of allogeneic HSCT recipients, methotrexate is an independent risk factor for OM [19,48,51,59]. The antiproliferative effects of methotrexate may be responsible for its role in OM incidence and severity [48,65]. Several strategies have been proposed to reduce OM risk in patients who require GVHD prophylaxis. One study found that in patients receiving methotrexate for GVHD prophylaxis in allogeneic HSCT, folinic acid, a folate derivative, demonstrated some benefits in reducing OM incidence [52]. While folinic acid has some prophylactic value, close monitoring of patients using methotrexate for signs of OM is still necessary [55]. Another study suggested that palifermin may have some protective value against severe OM in patients requiring methotrexate for allogeneic HSCT; however, there have not been enough studies in this specific population to make a recommendation [48]. Using a lower dose of methotrexate or a mycophenolate mofetil-based prophylactic regimen is an acceptable alternative to high-dose methotrexate that may reduce the risk of OM [6,51].

Patient-related risk factors such as an age < 40 years, renal dysfunction, female sex, as well as functional, immune, and nutritional status, contribute to OM risk. Several papers included in our study determined poor renal function to be associated with increased risk of OM in HSCT patients [38,43,47,50,55]. Multiple myeloma (MM) patients commonly experience some degree of kidney failure that may or may not require dialysis throughout their disease, and the standard of care for MM involves an autologous stem cell transplant [55,66,67]. Melphalan is a renally excreted conditioning agent that is commonly used in patients with MM undergoing HSCT. One study suggests that the association between poor renal function and toxicities may be due to current melphalan dosing procedures that are based on body surface area [38]. This strategy is not standardized and may be leading to overdoses of toxic chemotherapeutic agents due to reduced drug clearance in patients with renal dysfunction [38,55,68]. Physicians may want to consider modifying melphalan doses, optimizing kidney function before initiation of conditioning, or using prophylactic agents in patients who demonstrate renal disease [43,50]. HSCT therapy should not be withheld from patients with renal dysfunction in an attempt to minimize adverse effects because ultimately, this therapy will help their disease progression, similarly to patients with normal renal function, and may even improve overall kidney function [55,69].

Malnutrition impairs mucosal barrier function and weakens immune responses, making the oral epithelium more susceptible to injury from chemotherapy. Deficiencies in key nutrients can delay tissue repair and exacerbate inflammation, increasing the risk and severity of OM in HSCT patients [54]. Among these, magnesium deficiency (hypomagnesemia) has been shown to be a likely contributing factor for OM. Magnesium plays a critical role in epithelial regeneration, cellular proliferation, and immune modulation [70]. Low magnesium levels can impair mucosal healing and enhance inflammatory responses through cytokine activation and oxidative stress, compounding the effects of chemoradiotherapy. In HSCT patients, nephrotoxic medications, such as cisplatin, amphotericin B, and calcineurin inhibitors, commonly lead to renal magnesium wasting, placing patients at higher risk [71]. This also demonstrates another possible mechanism by which renal dysfunction may be associated with OM. Interestingly, however, Merve Savaş et al. (2024) found that hypomagnesemia conferred a protective effect on HSCT patients (HR = 0.380 [95% CI: 1.080–2.313]), a result which the authors acknowledged was contradictory to other known data and should thus be interpreted cautiously [56]. Conflicting associations between magnesium and transplant outcomes demonstrate the need for additional studies to understand the interplay between magnesium and endothelial function, particularly in the complex environment between donor and host cells in allogenic HSCT [56]. The study by Khosroshahi et al. (2023) revealed that malnourished patients had a higher incidence of OM compared to their well-nourished counterparts in leukemia patients undergoing allogeneic HSCT [54,61]. This underscores the importance of early nutritional assessment and intervention to improve patient outcomes during HSCT.

In our analysis, female sex emerged as a potential risk factor for the development of severe OM, consistent with previous studies [17,36,39,43,46,53,55,62]. Although the underlying mechanisms are not fully elucidated, there are several biologically plausible explanations. Women may experience increased mucosal toxicity due to sex-based differences in pharmacokinetics and drug metabolism, resulting in higher systemic exposure to chemotherapy agents such as melphalan or cytarabine [19,72]. In addition, hormonal influences, particularly the pro-inflammatory effects of estrogen, may heighten mucosal sensitivity and delay epithelial recovery following cytotoxic injury [19,73]. This association has also been suggested in genomic studies, where sex can potentially modify the expression of risk-related genetic polymorphisms [18]. While not universally observed across all cohorts, these findings support the inclusion of sex as a variable in future mucositis risk stratification tools and personalized prevention protocols.

Younger age (<40 years) has also been associated with increased risk of OM in HSCT recipients, as noted in Kashiwazaki et al. (2012) (OR = 5.6 [95% CI: 1.9–16.5]) [40]. Younger individuals generally exhibit higher rates of basal epithelial cell turnover, making their oral mucosa more vulnerable to cytotoxic injury from chemotherapy or TBI [40]. Moreover, patients under 40 are more likely to undergo damaging MAC regimens [6]. There is also the possibility of more accurate symptom reporting or greater mucosal sensitivity in younger patients, contributing to higher rates of OM detection. These findings underscore the need for individualized mucositis risk stratification based not only on frailty and organ function, but on age-related epithelial dynamics and treatment intensity as well.

In addition to demographic and nutritional factors, both functional and immunologic status appear to influence the risk of OM in HSCT patients. Blijlevens et al. (2008) reported that patients with poor functional performance (ECOG ≥ 2) experience higher rates of severe OM (OR = 1.8 [95% CI: 1.1–2.8]), likely due to impaired oral care, comorbidities, and limited capacity for mucosal repair [33]. Similarly, immune competence at baseline has emerged as a significant predictor of OM severity. Lee et al. (2018) identified that the presence of CD3^+^CD4^+^CD161^+^ T cells was associated with a markedly lower risk of OM (RR = 0.19 [95% CI: 0.04–0.73]), underscoring the protective role of mucosal-associated T cell subsets in epithelial healing and inflammatory regulation [43]. These findings support the integration of functional status and immune profiling into OM risk models, which may help identify high-risk patients and optimize the timing of prophylactic interventions.

Our study has several limitations. The nature of a systematic review prevents us from establishing a causal relationship between the risk factors identified and OM. The results from our study demonstrate the need for prospective studies to further evaluate risk factors of OM and the risk of further infection in cancer patients undergoing stem cell transplants. Additionally, while we ensured that included studies used clinically validated tools to confirm the diagnosis of OM and grade its severity, we cannot exclude the possibility of concomitant GI mucositis within these patients as a possible confounder for the relationship between OM and infection. Our study also possesses notable strengths. Introduction of bias is limited through our use of validated bias assessment tools and predetermined quality thresholds. This ensured that only studies with minimal potential for bias were included in our study. We were able to include 34 studies in our project, 4 of which were included in our meta-analysis. The number of studies served as a significant reservoir of data from which we could analyze factors associated with OM.

## 5. Conclusions

OM is a common, significant, and potentially dangerous consequence of hematopoietic stem cell treatment that may influence length of stay during transplant and subsequent care for patients with cancer [1,2]. Our study demonstrated an increased risk of serious, systemic infectious complications in patients with OM. Analysis of risk factors identified several patient-related factors, laboratory results, and features of conditioning regimens that are associated with increased risk of OM. Knowledge of OM risk factors for HSCT recipients with cancer could lead to the identification of high-risk individuals, a reduction in OM incidence, and protection of an immunocompromised population from subsequent life-threatening systemic infections. Preventive strategies such as oral cryotherapy, folinic acid, palifermin, and reduced-intensity conditioning regimens appear promising in reducing the risk of OM in select patient populations. Our findings indicate the necessity for more randomized controlled trials to evaluate the comparative effectiveness of these interventions and to explore emerging strategies such as targeted modulation of the oral microbiome and therapeutic approaches to OM.

Limitations: Readers are cautioned due to the following limitations. First, although numerous researchers participated in the data extraction and quality assessment, differences were reconciled through discussion and consensus; however, a formal evaluation of inter-rater reliability was not performed. This may create a risk of observer bias, as the uniformity of assessments among researchers cannot be objectively assessed. Nonetheless, validation was achieved through numerous researchers and subsequent consultation with the senior author, conducted via a blinded approach in which each researcher was unaware of the findings of others, ultimately resolved through conversation with the team and further aligning with the research aims, inclusion, and exclusion criteria. Second, an I-squared (I^2^) value of 0% in a meta-analysis indicates minimal statistical heterogeneity, suggesting that the variation in effect sizes across studies is negligible and largely attributable to random variation. However, this may also occur despite considerable clinical variability among trials, due to factors including sampling error, insufficient statistical power, inconsistent effect directions, misleading I^2^ interpretations, and the absence of outlier studies. While we ensured that included studies used clinically validated tools to confirm the diagnosis of OM and grade its severity, we cannot exclude the possibility of concomitant gastrointestinal mucositis within these patients. Third, the oral mucosa often shows earlier and more visible signs of injury compared to the intestinal mucosa. We acknowledge the competing risks between oral and gut-derived mucosal damage, and most studies did not distinguish the source of bloodstream pathogens or perform species-level comparisons. Future research incorporating metagenomic sequencing and microbial source tracking can define the contribution of site-specific mucosal injury to post-transplant infections. We did not explore this pathophysiologic hypothesis further, as it did not appear to be within the scope of this study.

## Figures and Tables

**Figure 1 cancers-17-02657-f001:**
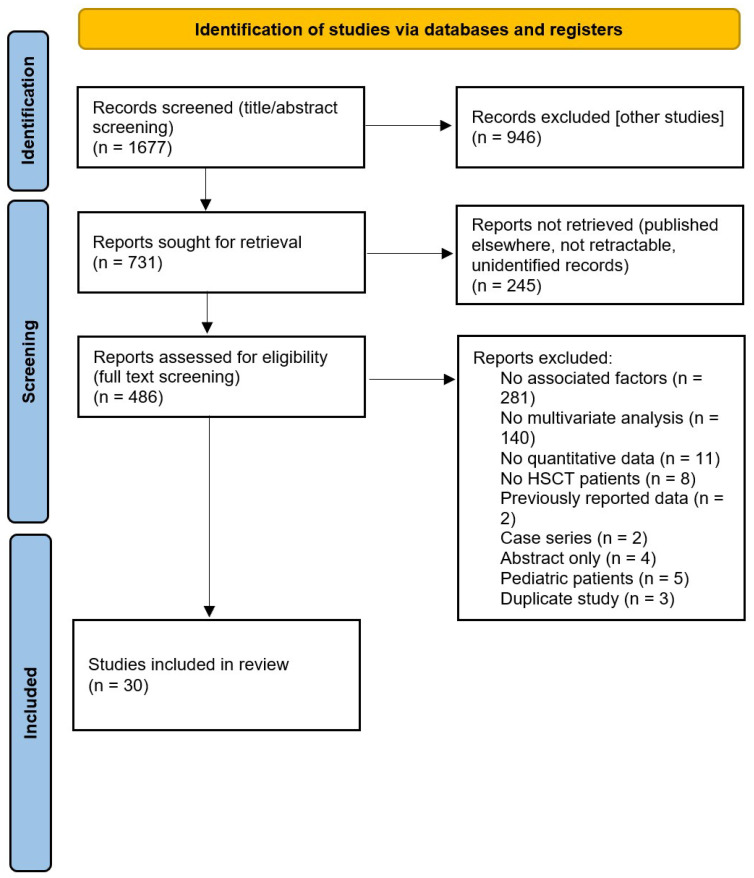
PRISMA flow chart for OM risk factor analysis [23].

**Figure 2 cancers-17-02657-f002:**
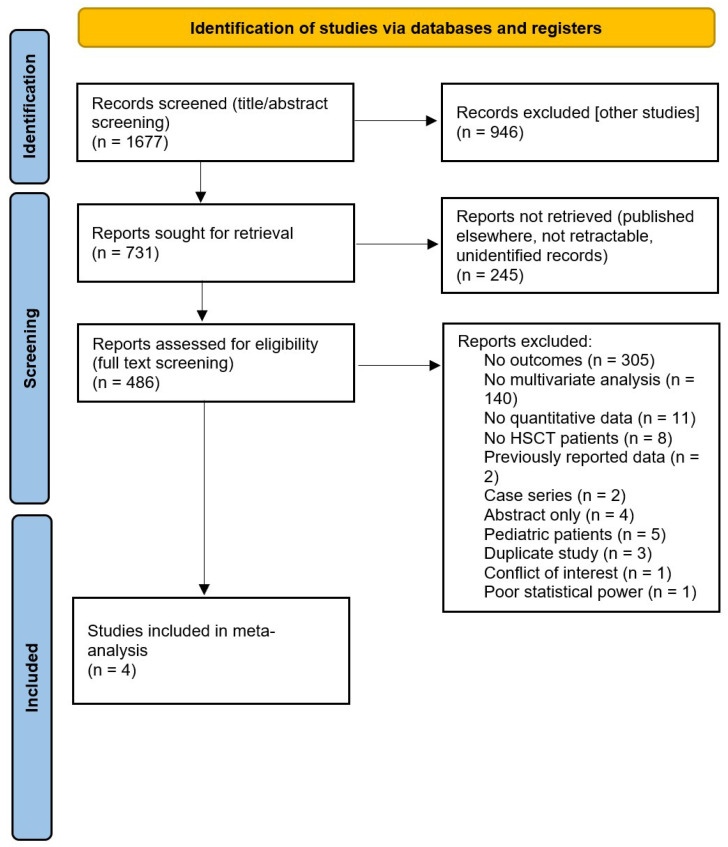
PRISMA flow chart for OM infectious complications [23].

**Figure 3 cancers-17-02657-f003:**
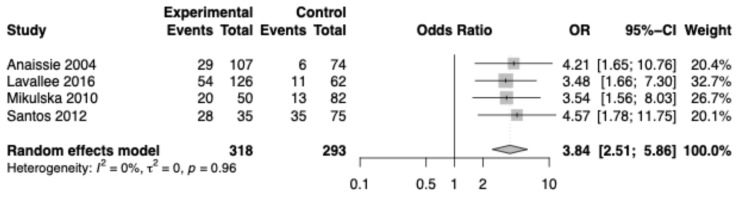
Meta-analysis [26,28,29,30].

**Table 1 cancers-17-02657-t001:** Characteristics of studies included in meta-analysis.

Reference, Year	Country	Study Design	Clinical Setting	Patient Population	Oral Mucositis Grading	Infectious Complication
Annaissie et al., 2004 [26]	USA	Prospective cohort	University of Arkansas for Medical Sciences in Little Rock	Cancer patients undergoing HSCT (n = 190)	Not stated	RSV + with complications
Deveci et al., 2022 [27]	Turkey	Retrospective cohort	Medstar Antalya Hospital	Autologous and allogeneic HSCT recipients with hematologic malignancy (n = 210)	Not stated	Typhlitis
Lavallee et al., 2016 [28]	Canada	Case-control	A single hospital in Montreal	Allogeneic HSCT recipients with hematologic malignancy (n = 760)	NCI-CTCAE version 3.0	*Clostridium difficile* + blood culture
Mikulska et al., 2010 [29]	Italy	Case-control	HSCT Unit of San Martino Hospital in Genoa	Allogeneic HSCT recipients with hematologic malignancy (n = 306)	WHO OM grading	Enterococcus bacteremia
Santos et al., 2012 [30]	Brazil	Cross-sectional	University Hospital, Universidade Federal de Juiz de Fora (UFJF)	Autologous HSCT recipients with hematologic malignancy (n = 112)	Not stated	Infection with + blood culture

**Table 2 cancers-17-02657-t002:** Characteristics of included studies in risk factor analysis.

Study	Study Design	Intervention or Exposure	Comparison	Clinical Setting	Patients	Oral Mucositis Grading	Outcome
Altes et al., 2007 [31]	Prospective cohort	Iron overload (above 75th percentile for ferritin and transferrin saturation)	Below the 75th percentile for ferritin and transferrin saturation	Not stated	HSCT recipients (n = 81)	NCI-CTCAE version 2.0	OM (grades 0–4), bacteremia, and fever
Batlle et al., 2014 [32]	Retrospective cohort	Cryotherapy	No cryotherapy	Single center	MM, NHL, or HL patients who underwent autologous HSCT with HDM conditioning (n = 134)	WHO OM grading	OM (grades 1–2 and 3–4) incidence and duration
Blijlevens et al., 2008 [33]	Prospective cohort	HDM conditioning	BEAM conditioning	Twenty-five centers across 13 European countries	MM and NHL patients (n = 214)	WHO OM grading	OM and severe OM (grades 3–4) duration and incidence
Cho et al., 2017 [34]	RCT	6 h of cryotherapy	2 h of cryotherapy	The Ohio State University, Ohio	Autologous HSCT recipients with MM (n = 146)	WHO OM grading	OM (grades 0–1 and 2–3)
Cho et al., 2019 [35]	Retrospective cohort	Glutamine-supplemented total parenteral nutrition	Non-glutamine-supplemented total parenteral nutrition	Seoul National University Bundang Hospital	HSCT recipients (n = 91)	Not stated	Weight change, infections, complications (mucositis, neutropenia, GVHD), and 100-day mortality
Coleman et al., 2015 [18]	Retrospective cohort	Total therapy treatment protocols	Non-total therapy treatment protocols	Myeloma Institute for Research and Treatment, Arkansas	Caucasian MM patients treated with autologous HSCT and HDM (n = 972)	CTCAE version 4.0	OM (grades 0–1 and 2–4
Gebri et al., 2020 [36]	Retrospective cohort	Lymphoma (NHL and HL) diagnosis	MM diagnosis	Hematopoietic Transplantation Centre of the Clinical Centre of the University of Debrecen, Hungary	Autologous HSCT recipients with hematological malignancies (n = 192)	WHO OM grading scale	OM (grades 0–1 and 2–4)
Gori et al., 2007 [37]	RCT	Cryotherapy	No cryotherapy	Institute of Hematology and Medical Oncology at the University of Bologna	Allogeneic HSCT patients undergoing MAC and MTX-containing GVHD prophylaxis (n = 130)	WHO OM grading	Severe OM (grades 3–4) incidence
Grazziutti et al., 2006 [38]	Retrospective cohort	200 mg melphalan dose	140 mg melphalan dose	Myeloma Institute for Research and Treatment, Arkansas	HDM and autologous HSCT recipients with MM (n = 381)	NCI-CTCAE version 2.0	OM and severe OM (grades 3–4) incidence
Hong et al., 2020 [39]	Prospective cohort	Presence of HSV-1/2 or Candida	Absence of HSV-1/2 or Candida	Seoul National University School of Dentistry	Patients with hematological malignancies receiving intensive chemotherapy or HSCT (n = 80)	WHO OM grading and NCI-CTCAE version 3.0	OM incidence (grades 1–4), subjective discomfort
Kashiwazaki et al., 2012 [40]	Retrospective cohort	RIC (FLUBU, FLUMEL)	Standard regimen (TBI, CY, with or without VP-16)	Stem Cell Transplantation Center of Hokkaido University Hospital	HSCT recipients (n = 130)	NCI-CTCAE version 3.0	OM incidence (grades 0–2 and 3–4)
Kawamura et al., 2013 [41]	Retrospective cohort	1000 mg acyclovir	200 mg acyclovir	Saitama Medical Center, Jichi Medical University	HSV-positive allogeneic HSCT recipients (n = 93)	Bearman scoring system	OM (grades 0–1 and 2–4) and HSV disease
Laheij et al., 2012 [42]	Prospective cohort	Presence of bacterial and Candida species	No presence of bacterial and Candida species	Leiden University Medical Center	HSCT patients with hematological malignancies (n = 49)	WHO OM grading	Ulcerative OM (grades 2–4) incidence
Lee et al., 2018 [43]	Prospective cohort	CD161 + T cells > 3.72%	CD161 + T cells ≤ 3.72%	One hospital in Korea	Autologous HSCT recipients with MM (n = 108)	NCI-CTC	OM (grades 1–2 and 3–4), infection, and cytomegalovirus reactivation
Lee et al., 2020 [44]	Prospective cohort	Autologous HSCT patients	Healthy volunteers	Seoul National University School of Dentistry	Adults who received oral examination (n = 61)	NCI-CTCAE version 3.0 and OM assessment scale	OM incidence (grades 0–4), HSV-1 detection, *Candida* detection, bacterial diversity
Legert et al., 2015 [45]	Prospective cohort	MAC (BUCY or CYTBI)	RIC (FLUBU, FLUCY, FLUTBI, FLUCYTBI, or CY)	Karolinska University Hospital, Huddinge, Sweden	Patients scheduled for HSCT (n = 77)	WHO OM grading	OM (grades 1–2 and 3–4) and serum and gingival crevicular fluid cytokine levels
Legert et al., 2021 [46]	RCT	Tacrolimus/Sirolimus	Cyclosporine/Methotrexate (standard regimen)	Two centers in Stockholm, Sweden, and Turku, Finland	Patients scheduled to receive allogeneic HSCT (n = 215)	OM assessment scale and WHO OM grading scale	NIH grade II-IV GVHD, OM (grades 0–1 and 2–4)
Nath et al., 2016 [47]	Prospective cohort	High melphalan dose (≥12.84 mg/hr)	Low melphalan dose (<12.84 mg/hr)	Six hospitals in the Autologous Working Party of BMT Network NSW, Australia	Autologous HSCT and HDM (n = 114)	NCI-CTCAE version 3.0	Severe OM (grades 3–4), time to progression, progression-free survival, and overall survival
Nguyen et al., 2015 [48]	Retrospective cohort	Palifermin	Historical control (no palifermin instituted)	City of Hope National Medical Center	Allogeneic HSCT recipients with hematological malignancies conditioned with full TBI and etoposide (n = 129)	NCI-CTCAE version 2.0	OM (grades 1–2 and 3–4) incidence
Rocha et al., 2009 [49]	Retrospective cohort	Presence of genetic polymorphisms	Absence of genetic polymorphisms	Hospital Saint Louis	Allogeneic HSCT recipients with leukemia (n = 107)	Research grading system	OM, hemorrhagic cystitis, liver toxicity, veno-occlusive disease, GVHD, and mortality
Salvador et al., 2005 [50]	Retrospective cohort	Primary prevention (before symptomatic OM)	Secondary prevention (after symptomatic OM)	University hospital in southern Ontario, Canada	Autologous HSCT recipients with MM, HL, or NHL (n = 140)	WHO OM grading	OM (grades 0–1 and 2–4) onset, incidence, and duration
Shouval et al., 2019 [51]	Retrospective cohort	MAC (BEAM, BUCY, CYTBI, FLUBU4, THIOFLUBU3)	RIC (FLAMZA, FLUBU2, FLUCYTBI, THIOFLUBU2) and Reduced Toxicity Conditioning (fludarabine and treosulfan)	Chaim Sheba Medical Center in Tel-Hashomer, Israel	Allogeneic HSCT recipients with hematological malignancies (n = 115)	CTCAE version 4.0	OM (grades 0–1 and 2–4)
Sugita et al., 2012 [52]	Retrospective cohort	Folinic acid (administered to high-risk patients)	No folinic acid	Hokkaido University Hospital, Japan	Allogeneic HSCT and MTX recipients (n = 141)	NCI-CTCAE version 3.0	OM (grades 1–2 and 3–4) incidence
Valeh et al., 2018 [53]	Prospective cohort	Allogeneic HSCT	Autologous HSCT	Hematology-Oncology and Stem Cell Transplantation Research Centre, Shariati Hospital, Tehran University of Medical Sciences	HSCT recipients (n = 173)	WHO OM grading	OM (grades 1–2 and 3–4) incidence and duration
Ursu et al., 2023 [55]	Retrospective cohort	Chronic kidney disease (Creatinine clearance less than 60 mL/min)	No chronic kidney disease (creatinine clearance greater than 60 mL/min)	Allegheny Health Network Cancer Institute	MM, who underwent autologous HSCT (n = 124)	NCI-CTCAE version 5.0	OM (grades 3 or 4) incidence
Khosroshahi et al., 2023 [54]	Prospective cohort	Presence of malnutrition based on GLIM criteria	Absence of malnutrition based on GLIM criteria	Hematology Center of Shariati Hospital in Tehran, Iran	Allogeneic HSCT recipients (n = 98)	WHO OM grading	OM (grades 2–4) incidence
Saori Oku et al., 2023 [59]	Retrospective cohort	Oral cryotherapy	No oral cryotherapy	Kyushu University Hospital, Japan	Allogeneic HSCT recipients (n = 78)	NCI-CTCAE version 3.0	OM (grades 1–3) incidence and duration
Wong et al., 2022 [57]	Prospective cohort	Presence of increasing inflammatory cytokines (TNF-α, IL-6, and IL-1β) in saliva and plasma	Absence of inflammatory cytokines in saliva and plasma	Ampang Hospital, Malaysia	Autologous HSCT recipients (n = 142)	WHO OM grading	OM (grades 1–4) incidence and duration
Lachance et al., 2023 [60]	Retrospective cohort	Bendamustine-based conditioning regimen	Carmustine-based conditioning regimen	Maisonneuve-Rosemont Hospital in Montreal, Quebec, Canada	Autologous HSCT recipients (n = 227)	Not stated	OM (grades 1–4) incidence
Merve Savaş et al., 2024 [56]	Retrospective cohort	Hypomagenesmia	Normal magnesium levels	Gazi University, Department of Hematology, Ankara, Turkey	Allogeneic HSCT recipients (n = 340)	NCI-CTCAE version 4.0	OM (grades 1–4) incidence

**Table 3 cancers-17-02657-t003:** Risk factor analysis for oral mucositis in cancer patients undergoing HSCT.

Variable	Study	Risk Estimate for OM
Baseline patient characteristics		
Age	Kashiwazaki et al., 2012 [40]	Age < 40 years: OR = 5.6 [1.9–16.5]
Sex	Garming Legert et al., 2021 [46] Gebri et al., 2020 [36] Lee et al., 2018 [43] Valeh et al., 2018 [53] Hong et al., 2020 [39] Ursu et al., 2023 [55]	Female sex: OR = 2.50 [1.15–5.42] Female sex: OR = 2.301 [1.124–4.714] Female sex: RR = 6.39 [1.74–29.71] Female sex: OR = 2.33 Female sex: OR = 0.221 [0.093–0.52] Female sex: OR = 4.2 [1.1–16.4]
HSV-1 presence	Hong et al., 2020 [39] Lee et al., 2020 [43]	OR = 7.660 [2.762–21.242] OR = 3.668 [1.512–8.895]
Renal function	Nath et al., 2016 [47] Lee et al., 2018 [43] Salvador et al., 2005 [50] Grazziutti et al., 2006 [38] Ursu et al., 2023 [55]	Beta-2 microglobulin: HR = 1.257 [1.035–1.528] GFR: RR = 0.98 [0.97–1.00] Peak Cr: Beta coefficient = 0.0283 Serum Cr: OR = 1.581 [1.080–2.313] CKD: OR = 8.2 [1.4–47.2]
Functional status	Blijlevens et al., 2008 [33]	ECOG performance: OR = 1.8 [1.1–2.8]
Immune status	Lee et al., 2018 [43]	Presence of CD3+CD4+CD161+ cells: RR = 0.19 [0.04–0.73]
Nutritional status	Khosroshahi et al., 2023 [54]	Presence of malnutrition based on GLIM criteria: OR = 1.39 [0.45–4.27]
Genetics	Rocha et al., 2009 [49] Coleman et al., 2015 [18]	CYP2B6*4 polymorphism: OR = 3.03 [1.37–6.73] CPEB1/LINC00692 (3p24.2) rs1426765 AA genotype: OR = 0.45 [0.32–0.65] FBN2 (5q23-q31) rs10072361 AA genotype: OR = 1.80 [1.29–2.51] FBN2 (5q23-q31) rs10072361 AG genotype: OR = 6.42 [2.16–19.07] FBN2 (5q23-q31) rs10072361 GG genotype: OR = 3.56 [1.18–10.81] ALDH1A1 (9q21.13) rs1469167 AA genotype: OR = 0.36 [0.22–0.58] DMTRA1/FLJ35282 (9p21.3) rs62572481 CC genotype: OR = 0.32 [0.18–0.58] DMTRA1/FLJ35282 (9p21.3) rs62572531 TT genotype: OR = 3.26 [1.81–5.84] MMP13 (11q22.3) rs1940228 AA genotype: OR = 0.27 [0.13–0.56] MMP13 (11q22.3) rs948695 AA genotype: OR = 0.25 [0.12–0.49] JPH3 (16q24.3) rs4843257 AA genotype: OR = 1.55 [1.08–2.21] JPH3 (16q24.3) rs4843257 AG genotype: OR = 2.56 [1.70–3.84] JPH3 (16q24.3) rs4843257 GG genotype: OR = 1.66 [1.17–2.34] DHRS7C (17p13.1) rs11078818 AG genotype: OR = 2.58 [1.11–5.98] DHRS7C (17p13.1) rs11078818 GG genotype: OR = 1.88 [1.37–2.60] CEP192 (18p11.21) rs12606033 GG genotype: OR = 1.97 [1.41–2.77]
Laboratory results		
Ferritin level	Altes et al., 2007 [31]	RR = 3.4 [1.1–10]
Duration of neutropenia	Kashiwazaki et al., 2012 [40] Sugita et al., 2012 [52] Gebri et al., 2020 [36]	OR = 12.4 [1.4–109] OR = 4.78 [1.77–13.90] OR = 1.492 [1.228–1.813]
Oral microbiota	Laheij et al., 2012 [42]	Presence of *P. gingivalis* (non-keratinized mucosal involvement): beta coefficient = 3.36 Presence of *C. kefyr* (non-keratinized mucosal involvement): beta coefficient = 2.01 Load of *P. gingivalis* (non-keratinized mucosal involvement): beta coefficient = 1.37 Load of *C. kefyr* (non-keratinized mucosal involvement): beta coefficient = 2.056 Percentage of *P. gingivalis* (non-keratinized mucosal involvement): beta coefficient = 1.372 Percentage of *P. micra* (non-keratinized mucosal involvement): beta coefficient = 0.00 Percentage of *F. nucleatum* (non-keratinized mucosal involvement): beta coefficient = 1.58 Percentage of *T. denticola* (non-keratinized mucosal involvement): beta coefficient = 0.87 Percentage of *C. glabrata* (non-keratinized mucosal involvement): beta coefficient = 3.49 Presence of *P. gingivalis* (keratinized mucosal involvement): beta coefficient = 4.38 Presence of *P. micra* (keratinized mucosal involvement): beta coefficient = 0.46 Load of *P. gingivalis* (keratinized mucosal involvement): beta coefficient = 0.75 Load of *C. kefyr* (keratinized mucosal involvement): beta coefficient = 1.83
Serum magnesium level	Merve Savaş et al., 2024 [56]	Serum magnesium less than 1.33 mg/dL: HR = 0.380 [0.161–0.896]
Inflammatory cytokines in saliva and plasma	Wong et al., 2022 [57]	Increase in plasma IL-6 by 10 pg/mL: OR = 1.01 [1.001–1.004] Increase in saliva IL-6 by 100 pg/mL: OR = 1.003 [1.001–1.004] Reduction in plasma TNF-⍺ by 10 pg/mL: OR = 0.91 [0.85–0.99]
Cancer treatment and conditioning regimens		
Chemotherapy	Salvador et al., 2005 [50] Batlle et al., 2014 [32]	NHL regimen vs. HL regimen: beta coefficient = 1.4712 ≥2 treatment lines before HSCT: OR = 3.103 [1.035–9.300]
HSCT modality	Cho et al., 2019 [35]	Autologous HSCT: beta coefficient = 0.38
Conditioning regimen	Blijlevens et al., 2008 [33] Cho et al., 2017 [34] Grazziutti et al., 2006 [38] Nath et al., 2016 [47] Batlle et al., 2014 [32] Gori et al., 2007 [37] Cho et al., 2019 [35] Kawamura et al., 2013 [41] Garming Legert et al., 2015 [45] Garming Legert et al., 2021 [46] Shouval et al., 2019 [51] Saori Oku et al., 2023 [59] Wong et al., 2022 [57] Lachance et al., 2022 [60]	HDM: OR = 2.6 [1.6–4.4] High dose carmustine: OR = 1.9 [1.3–2.6] HDM: RR = 1.21 [1.04–1.41] HDM: OR = 1.595 [1.065–2.389] HDM: HR = 1.213 [1.064–1.382] Use of BEAM: OR = 3.633 [1.181–11.176] TBI: RR = 3.2 [1.4–7.6] MAC: Beta coefficient = 1.11 [0.295–4.18] MAC: OR = 7.22 [2.66–19.50] MAC: OR = 1.37 [1.03–1.82] Reduced intensity conditioning: OR = 0.18 [0.06–0.56] Reduced intensity conditioning: RR = 0.04 [0.01–0.17] HDM: OR = 3.82 [1.085–13.46] BEAM or busulphan based regimen: OR = 9.2 [1.16–72.9] Bendamustine based conditioning regimen: HR = 2.946 [1.19–7.27]
Methotrexate use	Nguyen et al., 2015 [48] Shouval et al., 2019 [51] Saori Oku et al., 2023 [59]	OR = 3.21 [1.38–7.46] RR = 3.53 [ 1.15–10.81] OR = 7.61 [2.41–23.97]
OM prophylaxis		
Folinic acid	Sugita et al., 2012 [52] Gori et al., 2007 [37]	Use of folinic acid: OR = 0.13 [0.04–0.73] Lack of folinic acid: RR = 2.6 [1.2–5.7]
Cryotherapy	Batlle et al., 2014 [32]	Lack of cryotherapy: OR = 8.345 [3.342–20.837]
Prophylaxis	Valeh et al., 2018 [53] Salvador et al., 2005 [50]	Use of prophylaxis: OR = 0.47 Primary prevention vs. secondary prevention: Beta coefficient = 0.9356

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
