# Peer review of "Elevated Likelihood of Infectious Complications Related to Oral Mucositis After Hematopoietic Stem Cell Transplantation: A Systematic Review and Meta-Analysis of Outcomes and Risk Factors"

_cancers, 2025, doi:10.3390/cancers17162657_

Round 1

Reviewer 1 Report

Comments and Suggestions for Authors

1. The introduction can be rewritten to focus more specifically on the objective of the study. Broad discussions can be avoided which stray from the main subject. Begin by briefly setting the context with relevant background information, then identify the research gap. Conclusion can be clear and focused with the study's aim. Incorporate keywords judiciously to ensure relevance.
2. In the inclusion criteria section of for analysis of infectious complications. the authors need to confirm whether the observational and randomized controlled studies assessed for quality or risk of bias prior to inclusion
3. The authors need to ensure that the references used in the manuscript are directly relevant to your study and limit the total number of citations and avoid citing unrelated reviews or articles.
4. The authors need to revise the conclusion of the manuscript to present a concise summary of your findings, also, highlight how your study contributes to the field and meets its objectives, avoiding repetition or elaboration.
5. The authors should clarify whether inter-rater reliability was assessed during scoring. Also, the thresholds used should be justified or referenced.
6. Use abbreviations only for terms that used frequently throughout the manuscript. Avoid introducing abbreviations for terms mentioned once or twice. If an abbreviation is used, then ensure it is used consistently throughout the text.
7. The authors could briefly mention which preventive or therapeutic strategies appear most promising based on their findings along with future direction.

Comments on the Quality of English Language

Not applicable

Author Response

Point by Point Reply to the comments.

July 28th

Manuscript ID: cancers-3760499

Title:

Elevated Likelihood of Infectious Complications Related to Oral Mucositis after Hematopoietic Stem Cell Transplantation: A Systematic Review and Meta-analysis of Outcomes and Risk Factors.

Corresponding Author: Poolakkad S. Satheeshkumar.

Reply:

Dear Editor,

Thank you for forwarding the editorial comments for our paper cited above.

We appreciate the thoroughness of the reviewers and are pleased to respond to their suggestions.

Specifically, we have made the following modifications:

Reviewer 1 comments.

  1. The introduction can be rewritten to focus more specifically on the objective of the Broad discussions can be avoided which stray from the main subject. Begin by briefly setting the context with relevant background information, then identify the research gap. The conclusion can be clear and focused with the study’s aim. Incorporate keywords judiciously to ensure relevance.

ANSWER: Thank you for this comment. Kindly note that both the introduction and conclusion were revised to be more succinct with a clear focus on the study objectives.

Introduction section:

Oral and oropharyngeal mucositis (OM) is a well-known consequence of cancer therapy, particularly hematopoietic stem cell transplantation (HSCT), and is associated with pain, nutritional deficiencies, dehydration, respiratory distress, treatment delays, and increased risk of febrile neutropenia and systemic infections 1,2,3] The preparatory or conditioning regimen for HSCT, comprised of high-dose chemotherapy with or without total body irradiation (TBI), is designed to eradicate residual malignant cells and immunosuppress the host to allow engraftment, and itself can harm the mucosal layer and other highly replicative cells in the body.[4] Following the infusion of stem cells, patients enter a prolonged aplastic period where granulocyte recovery is delayed, leaving them susceptible to infection and mucosal injury. Engraftment is marked by the return of neutrophil counts, typically around day +14 to +21, though this timing can vary depending on the graft source and patient-specific factors.[4] Hospitalization during this phase is often prolonged, particularly in patients with delayed engraftment, graft-versus-host disease (GVHD), or infectious complications.[4] Notably, both pediatric and adult populations undergo HSCT but elderly recipients, whose numbers have risen significantly in recent years alongside rising rates of HSCT treatments, are often at increased risk due to comorbidities and reduced mucosal regenerative capacity.[5,6]

According to recent studies, 80% of patients undergoing HSCT are expected to develop OM, 35%–75% of those with autologous transplants and 75%–100% of those with allogeneic transplants.[2,7,8] These rates aresignificantly higher than in patients receiving chemotherapy alone.[2] Not only is OM prevalent, but it tends to occur in advanced forms, with 42-60% of HSCT patients developing severe OM.[2,9] This may lead to chemotherapy dose reduction or discontinuation, with negative consequences for patient outcomes.[2,9] The oral cavity, particularly when compromised by mucositis, serves as a reservoir for pathogenic organisms capable of translocating into systemic circulation, especially in immunocompromised populations such as hematologic malignancy patients.[10,11] Studies have shown that bloodstream infections are estimated to affect about 5–10% of autologous and 20–30% of allogeneic HSCT recipients, though this figure can change on a center-by-center and patient-by-patient basis.[12] For example, some recent studies have even shown rates as high as 77% for autologous transplant and 48% for allogeneic transplant.[13,14] Given that OM is a predictable, dose-limiting toxicity in patients receiving high-dose chemotherapy or HSCT,[1,2] identifying its role in systemic infection is critical for developing preventative strategies.

Unfortunately, methods to prevent OM remain limited, and most commonly involve oral cryotherapy, growth factors, and benzydamine. Once an individual develops OM, there is no cure and treatment is focused on managing pain with analgesics and low-level laser therapy if available.[1] Poor oral health[15,16], smoking[15,17], genetic polymorphisms[18], and high intensity conditioning regimens[2,15,17,19] are consideredrisk factors for OM in HSCT recipients. However, the majority of information is collected from randomized controlled trials (RCT), in which females and minorities are underrepresented,[20,21] hence compiling results from observational studies to obtain real-world evidence is needed at any cost.  A 2015 study by Chaudhry et al. analyzed risk of OM in allogeneic HSCT patients in terms of regimen intensity and GHVD prophylaxis with or without methotrexate.[19] A subsequent study by Wardill et al. in 2020 looked more broadly at OM risk factors in patients receiving cancer therapy, not only those undergoing HSCT.[17] It assessed not only treatment modalities but patient characteristics including genetic polymorphisms and demographic/lifestyle elements. We look to expand these areas of knowledge by further examining patient and clinical risk factors associated with OM in HSCT patients, as well as infectious outcomes.  In doing so, we hope to identify higher-risk individuals and ultimately reduce incidence through earlier detection and employment of prophylactic strategies. Our objectives are thus twofold:

(1) Identify risk factors associated with OM across HSCT modalities from multivariate analyses.

(2) Quantify the risk of infectious complications associated with OM.

Conclusion section:

OM is a common, significant, and potentially dangerous consequence of hematopoietic stem cell treatment that may influence length of stay during transplant and subsequent care for patients with cancer.[1,2]Our study demonstrated an increased risk of serious, systemic infectious complications in patients with OM. Analysis of risk factors identified several patient-related factors, laboratory results, and features of conditioning regimens that are associated with increased risk of OM. Knowledge of OM risk factors for HSCT recipients with cancer could lead to identification of high-risk individuals, reduction of OM incidence, and protection of an immunocompromised population from subsequent life-threatening systemic infections. Preventive strategies such as oral cryotherapy, folinic acid, palifermin, and reduced-intensity conditioning regimens appear promising in reducing the risk of OM in select patient populations. Our findings indicate the necessity for more randomized controlled trials to evaluate the comparative effectiveness of these interventions and to explore emerging strategies such as targeted modulation of the oral microbiome and therapeutic approaches to OM.

  1. In the inclusion criteria section of for analysis of infectious complications. the authors
    need to confirm whether the observational and randomized controlled studies assessed
    for quality or risk of bias prior to inclusion.

ANSWER: Thank you for this comment. Studies needed to achieve minimum bias ratings as detailed in results section and recorded in appendix 2-4. Bias assessments were performed after papers meet inclusion and exclusion criteria listed in methodology.

  1. The authors need to ensure that the references used in the manuscript are directly relevant to your study and limit the total number of citations and avoid citing unrelated reviews or articles.

ANSWER: Thank you for this comment. References were reviewed for relevance and condensed appropriately.

  1. The authors need to revise the conclusion of the manuscript to present a concise summary of your findings, also, highlight how your study contributes to the field and meets its objectives, avoiding repetition or elaboration.

ANSWER: Thank you for this comment. The conclusion was revised to succinctly present key findings that fulfilled the objectives of our study.

  1. The authors should clarify whether inter-rater reliability was assessed during scoring. Also, the thresholds used should be justified or referenced.

ANSWER: Thank you for this comment. We have provided a limitation section.

Limitations: Although numerous researchers participated in the data extraction and quality assessment procedure, differences were reconciled through discussion and consensus; however, a formal evaluation of inter-rater reliability was not performed. This may create a risk of observer bias, as the uniformity of assessments among researchers cannot be objectively assessed. Nonetheless, validation was achieved through numerous researchers and subsequent consultation with the senior author, conducted via a blinded approach in which each researcher was unaware of the findings of others, ultimately resolved through conversation with the team and further aligning with the research aims, inclusion and exclusion criteria. Subsequent systematic studies on this subject should contemplate the inclusion of formal inter-rater reliability evaluations to augment the objectivity and reproducibility of the review process.

  1. Use abbreviations only for terms that used frequently throughout the manuscript. Avoid introducing abbreviations for terms mentioned once or twice. If an abbreviation is used, then ensure it is used consistently throughout the text.

ANSWER: Thank you for this comment. We ensured that only relevant abbreviations were listed, and once identified, the abbreviated terms were used frequently throughout the paper.

  1. The authors could briefly mention which preventive or therapeutic strategies appear most promising based on their findings along with future direction.

ANSWER: Thank you for this comment. Preventative and therapeutic strategies identified in the studies were reviewed in detail in the discussion section. In the conclusion section, we added a brief statement highlighting the most promising strategies and a statement with future directions for studying these strategies.

Reviewer 2 Report

Comments and Suggestions for Authors

This is a well-executed systematic review and meta-analysis examining risk factors for oral mucositis (OM) and its association with infectious complications in patients undergoing HSCT. The study is timely and clinically relevant, employing robust methodology (PRISMA, Newcastle-Ottawa/Cochrane assessments) and providing meaningful pooled estimates. There are some minor points to address:

  • Why statistical heterogeneity was low (I² = 0%) despite clinical variability across included studies.
  • Clarify whether infection outcomes were limited to microbiologically documented or also included clinically suspected cases.
  • Revisit the interpretation of the unexpected protective association of hypomagnesemia, emphasizing the need for cautious interpretation.

Author Response

Point by Point Reply to the comments.

July 28th

Manuscript ID: cancers-3760499

Title:

Elevated Likelihood of Infectious Complications Related to Oral Mucositis after Hematopoietic Stem Cell Transplantation: A Systematic Review and Meta-analysis of Outcomes and Risk Factors.

Corresponding Author: Poolakkad S. Satheeshkumar.

Reply:

Dear Editor,

Thank you for forwarding the editorial comments for our paper cited above.

We appreciate the thoroughness of the reviewers and are pleased to respond to their suggestions.

Specifically, we have made the following modifications:

Reviewer 2 comments

This is a well-executed systematic review and meta-analysis examining risk factors for oral mucositis (OM) and its association with infectious complications in patients undergoing HSCT. The study is timely and clinically relevant, employing robust methodology (PRISMA, Newcastle-Ottawa/Cochrane assessments) and providing meaningful pooled estimates.

Answers: We appreciate reviewers’ comments and inspiration to write the editing.

There are some minor points to address:

  1. Why statistical heterogeneity was low (I² = 0%) despite clinical variability across included studies?

ANSWER: Thank you for this comment. A low I-squared (I²) value (e.g., 0%) in a meta-analysis signifies minimal statistical heterogeneity, indicating that the diversity in effect sizes among studies is negligible and primarily due to random variation. Nonetheless, this may transpire despite significant clinical diversity among trials, attributable to various factors such as sampling error, inadequate statistical power, inconsistent direction of effects, misleading I² interpretation, and lack outlying studies. And a low I² statistic does not inherently indicate the absence of clinical variability; it merely suggests that the observed differences among studies are minimal enough to be attributed to chance, that the study design lacks adequate power to identify genuine differences, or that the effects are consistently directed. We have provided a limitation in the limitation section.

An I-squared (I²) value of 0% in a meta-analysis indicates minimal statistical heterogeneity, suggesting that the variation in effect sizes across studies is negligible and largely attributable to random variation. However, this may occur despite considerable clinical variability among trials, due to factors including sampling error, insufficient statistical power, inconsistent effect directions, misleading I² interpretations, and the absence of outlier studies.

  1. Clarify whether infection outcomes were limited to microbiologically documented or also included clinically suspected cases.

ANSWER: Thank you for this comment. Language was added to the results section to clarify that clinically defined and microbiologically defined systemic infectious outcomes were studied.

  1. Revisit the interpretation of the unexpected protective association of hypomagnesemia, emphasizing the need for cautious interpretation.

ANSWER: Thank you for this comment. The interpretation of association between hypomagnesemia and OM was reviewed and edited to provide caution.

Reviewer 3 Report

Comments and Suggestions for Authors

The authors seem to have done an exhaustive effort in producing this manuscript. It explores the relationship between oral mucositis induced by the conditioning regimens for hematopoietic stem cell transplantation and the increased risk of infectious complications. Overall, an interesting read.

However, I have a few queries: how do the authors dissect oral mucositis from the mucositis involving other parts of the gastro-intestinal system like small and large intestine to attribute the increased infections to breakage of oral mucosal barrier alone? These are competing risks. What about oral mucositis being a visible reflection of more extensive and consequential mucositis elsewhere in the GIT? The extent of damage from the same intensity of conditioning regimen to the stratified squamous epithelium of oral mucosa and the single layer columnar epithelial lining of the intestines may be very different. What would a comparison of oral derived microorganisms versus gut derived microorganisms isolated in blood cultures in post conditioning bacteremia look like?

Also, I do not find the relationship between resident microbiota and protection form or proneness to mucosal barrier integrity loss from varying intensity conditioning regimens and by extension, variability of resident microbiota and infectious complications explored.    

Author Response

Point by Point Reply to the comments.

July 28th

Manuscript ID: cancers-3760499

Title:

Elevated Likelihood of Infectious Complications Related to Oral Mucositis after Hematopoietic Stem Cell Transplantation: A Systematic Review and Meta-analysis of Outcomes and Risk Factors.

Corresponding Author: Poolakkad S. Satheeshkumar.

Reply:

Dear Editor,

Thank you for forwarding the editorial comments for our paper cited above.

We appreciate the thoroughness of the reviewers and are pleased to respond to their suggestions.

Specifically, we have made the following modifications:

Reviewer 3 comments

The authors seem to have done an exhaustive effort in producing this manuscript. It explores the relationship between oral mucositis induced by the conditioning regimens for hematopoietic stem cell transplantation and the increased risk of infectious complications. Overall, an interesting read.

Answers: We appreciate reviewer comments, and this inspired us to complete the editing.

However, I have a few queries:

  1. How do the authors dissect oral mucositis from the mucositis involving other parts of the gastro-intestinal system like small and large intestine to attribute the increased infections to breakage of oral mucosal barrier alone? These are competing risks. What about oral mucositis being a visible reflection of more extensive and consequential mucositis elsewhere in the GIT? The extent of damage from the same intensity of conditioning regimen to the stratified squamous epithelium of oral mucosa and the single layer columnar epithelial lining of the intestines may be very different. What would a comparison of oral derived microorganisms versus gut derived microorganisms isolated in blood cultures in post conditioning bacteremia look like? Also, I do not find the relationship between resident microbiota and protection form or proneness to mucosal barrier integrity loss from varying intensity conditioning regimens and by extension, variability of resident microbiota and infectious complications explored.  

ANSWER: Thank you for this comment. While we ensured included studies used clinically validated tools to confirm the diagnosis of OM and grade its severity, we cannot exclude the possibility of concomitant GI mucositis within these patients. This caveat was added as a limitation of our study. The oral mucosa, with its stratified squamous epithelium, often shows earlier and more visible signs of injury compared to columnar-lined intestinal mucosa. We acknowledge the competing risks between oral and gut-derived mucosal damage, and most studies did not distinguish the source of bloodstream pathogens or perform species-level comparisons. Future research incorporating metagenomic sequencing and microbial source tracking can define the contribution of site-specific mucosal injury to post-transplant infections. We did not explore this pathophysiologic hypothesis further, as it did not appear to be within the scope of this study.

Round 2

Reviewer 1 Report

Comments and Suggestions for Authors

The authors have carried out all suggestions

Author Response

Dear Reviewer 1, thank you for this comment.

Reviewer 3 Report

Comments and Suggestions for Authors

The oral mucosa, with its stratified squamous epithelium, often shows earlier and more visible signs of injury compared to columnar-lined intestinal mucosa.

it would be good to remove the above cancelled phrases.

Author Response

Point by Point Reply to the comments.

August 8th

Manuscript ID: cancers-3760499

Title:

Elevated Likelihood of Infectious Complications Related to Oral Mucositis after Hematopoietic Stem Cell Transplantation: A Systematic Review and Meta-analysis of Outcomes and Risk Factors.

Corresponding Author: Poolakkad S. Satheeshkumar.

Reply:

Dear Editor,

Thank you for forwarding the editorial comments for our paper cited above.

We appreciate the thoroughness of the reviewers and are pleased to respond to their suggestions.

Specifically, we have made the following modifications:

Reviewer 3 comments.

The oral mucosa, with its stratified squamous epithelium, often shows earlier and more visible signs of injury compared to columnar-lined intestinal mucosa. it would be good to remove the above cancelled phrases.

ANSWER: Thank you for this comment. We have added a limitation section

While we ensured included studies used clinically validated tools to confirm the diagnosis of OM and grade its severity, we cannot exclude the possibility of concomitant gastrointestinal mucositis within these patients. The oral mucosa often shows earlier, and more visible signs of injury compared to intestinal mucosa. We acknowledge the competing risks between oral and gut-derived mucosal damage, and most studies did not distinguish the source of bloodstream pathogens or perform species-level comparisons. Future research incorporating metagenomic sequencing and microbial source tracking can define the contribution of site-specific mucosal injury to post-transplant infections. We did not explore this pathophysiologic hypothesis further, as it did not appear to be within the scope of this study.
